# Genome-wide identification of lineage and locus specific variation associated with pneumococcal carriage duration

John A Lees[1]*, Nicholas J Croucher[2], David Goldblatt[3], François Nosten[4,5], Julian Parkhill[1], Claudia Turner[4,5‡], Paul Turner[4,5†‡], Stephen D Bentley[1†]*

[1]Infection Genomics, Wellcome Trust Sanger Institute, Hinxton, United Kingdom; [2]Department of Infectious Disease Epidemiology, St. Mary's Campus, Imperial College London, London, United Kingdom; [3]Institute of Child Health, University College London, London, United Kingdom; [4]Shoklo Malaria Research Unit, Mahidol-Oxford Tropical Medicine Research Unit, Faculty of Tropical Medicine, Mahidol University, Mae Sot, Thailand; [5]Centre for Tropical Medicine and Global Health, Nuffield Department of Medicine, University of Oxford, Oxford, United Kingdom

**Abstract** *Streptococcus pneumoniae* is a leading cause of invasive disease in infants, especially in low-income settings. Asymptomatic carriage in the nasopharynx is a prerequisite for disease, but variability in its duration is currently only understood at the serotype level. Here we developed a model to calculate the duration of carriage episodes from longitudinal swab data, and combined these results with whole genome sequence data. We estimated that pneumococcal genomic variation accounted for 63% of the phenotype variation, whereas the host traits considered here (age and previous carriage) accounted for less than 5%. We further partitioned this heritability into both lineage and locus effects, and quantified the amount attributable to the largest sources of variation in carriage duration: serotype (17%), drug-resistance (9%) and other significant locus effects (7%). A pan-genome-wide association study identified prophage sequences as being associated with decreased carriage duration independent of serotype, potentially by disruption of the competence mechanism. These findings support theoretical models of pneumococcal competition and antibiotic resistance.

DOI: https://doi.org/10.7554/eLife.26255.001

*For correspondence: jl11@ sanger.ac.uk (JAL); sdb@sanger. ac.uk (SDB)

[†]These authors contributed equally to this work

**Present address:** [‡]Cambodia-Oxford Medical Research Unit, Angkor Hospital for Children, Siem Reap, Cambodia

**Competing interests:** The authors declare that no competing interests exist.

## Introduction

*Streptococcus pneumoniae* is a human pathogen that can cause diseases such as pneumonia, otitis media and meningitis. Pneumococcal disease burden is highest in children (*O'Brien et al., 2009*). For disease to be caused pneumococci must first transmit to the host, colonise the nasopharynx and finally cross into a normally sterile site. The pneumococcus spends most of the transmission cycle in the nasopharynx, and so understanding and predicting the amount of time spent in this niche is critical for understanding this bacterium's epidemiology, and therefore controlling transmission (*Abdullahi et al., 2012a*; *Melegaro et al., 2007*).

The nasopharynx is a complex niche in which each pneumococcal genotype must tackle a wide range of factors including host immune defence (*McCool et al., 2002*), other bacterial species (*Pericone et al., 2000*), and other pneumococcal lineages (*Auranen et al., 2010*; *Cobey and Lipsitch, 2012*) in order to maintain the genotype's population. The average nasopharyngeal duration period is therefore affected by a large number of factors, which may, themselves, interact.

One factor that is known to strongly associate with carriage duration is serotype: as capsular polysaccharides are important in bacterial physiology and determining host immune response, different

**eLife digest** Microorganisms live in most parts of our body, including the inside of our nose. Most of the microbes are harmless and can even be beneficial to our health. However, some microbes can cause diseases – although they often go unnoticed, as our immune system can remove them before we show any symptoms. For example, the bacterium *Streptococcus pneumoniae* can cause diseases such as pneumonia and meningitis, but generally, it lives harmlessly in the nose, and is particularly common in children and the elderly.

The longer the bacteria live in the nose before being killed by the immune system, the more likely they are to be transmitted to another person. The amount of time it takes for the immune system to clear the bacteria depends on various factors, such as the age of the person or the bacterium's defense mechanism and its genetic material. A particularly important aspect is to what subtype, also known as serotype, a bacterium belongs to, which is characterized by differences in the structure of the sugar coating that surrounds the microbe. However, until now, it was not known how much each of these factors contributes.

Now, Lees et al. have developed a mathematical model to calculate how long the bacteria are carried in the nose before they are cleared away, and compared it with the genomic data of the bacteria. For this, over 14,000 nose swabs from almost 600 children were collected over a two-year period. In their model, Lees et al. calculated that the bacteria's genetics explained over 60% of the variability in survival time. They also found that the serotype was the most important individual factor that influenced how long a bacterium could survive. The age of the child was less important and only accounted for 5%. In addition, Lees et al. also found that when viruses infected some *S. pneumoniae*, the bacteria died sooner.

A next step will be to confirm the effect of a viral infection on the bacteria's survival time in a controlled model system, and also replicate the findings in separate population study.Understanding how long people can carry bacteria and transmit them to others may help to develop new vaccination or treatment strategies to control infections. Moreover, the discovery that viruses can negatively affect how long a bacterium lives, could motivate studies to investigate these findings further.

DOI: https://doi.org/10.7554/eLife.26255.002

serotypes have different clearance and acquisition rates (*Abdullahi et al., 2012a*; *Hill et al., 2010*; *Högberg et al., 2007*; *Melegaro et al., 2004*; *Turner et al., 2012*). Additionally, a range of other proteins have been identified as critical to the colonisation process (*Kadioglu et al., 2008*), some of which exhibit similar levels of diversity to the capsule polysaccharide synthesis locus (*Iannelli et al., 2002*; *Jedrzejas et al., 2001*). However, the overall and relative contributions of these sequence variations to carriage rate have not yet been characterised. In addition variation of pathogen protein sequence, accessory genes and interaction effects between genetic elements may also have as yet unknown effects on carriage duration.

Changes in average carriage duration have been shown to be linked with recombination rate (*Chaguza et al., 2016*), which has been found to correlate with antibiotic resistance (*Hanage et al., 2009*) and invasive potential (*Chaguza et al., 2016*). The carriage duration by different serotypes is widely used in models of pneumococcal epidemiology, and consequently is important in evaluating the efficacy of the pneumococcal conjugate vaccine (PCV) (*Melegaro et al., 2007*; *Weinberger et al., 2011*). Additionally, modelling work has proposed that if alleles exist which alter carriage duration, these explain the long standing puzzle of how antibiotic-resistant and sensitive strains stably coexist in the population (*Lehtinen et al., 2017*). Measurement of carriage duration and the analysis of its variance beyond the resolution of serotype will have important consequences for these models.

We sought to determine the overall importance of the pathogen genotype in carriage duration in a human population, and to identify and quantify the elements of the genome responsible for the variation in carriage duration. By combining epidemiological modelling of longitudinal swab data with and genome wide association study methods on the connected sequences (*Figure 1*), we made heritability estimates for carriage duration. We further partitioned the heritability into contributions

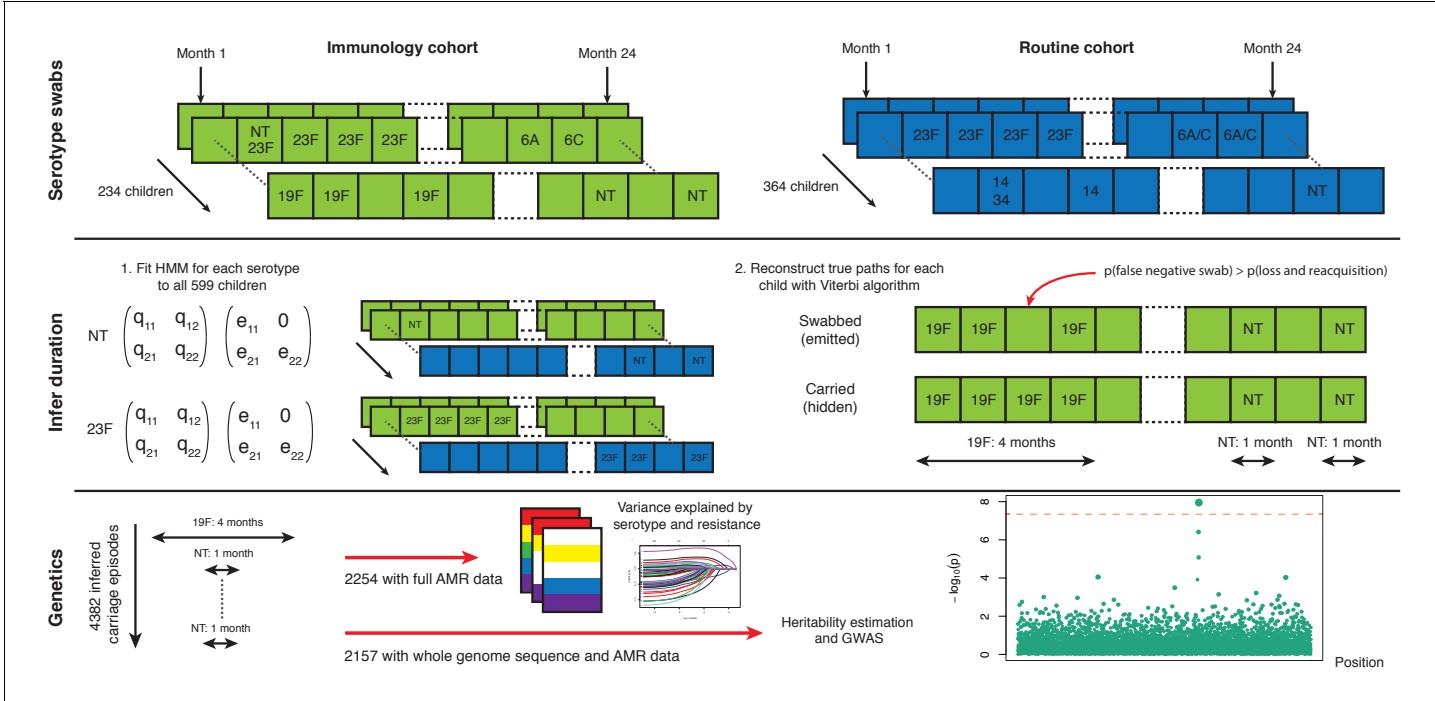

**Figure 1.** Swabbing and sequencing study design. We start with serotype swab data on 598 children from two cohorts, taken every month after birth for two years. For all samples we fitted the transition and emission probabilities of a continuous time hidden Markov model for each serotype. Then, for each child, we used these parameters were then used to infer the most likely carriage durations. We matched carriage episodes with resistance and genomic data for 2157 episodes to draw conclusions on the basis of variation in this epidemiological parameter.
DOI: https://doi.org/10.7554/eLife.26255.003

from lineage and locus effects (*Earle et al., 2016*) to quantify the variation caused by each individual factor.

## Results

### Ascertainment of carriage episode duration using epidemiological modelling

We first estimated carriage duration from longitudinal swab data available for the study population. For 598 unvaccinated children up to 24 swabs taken over a two year period were available, an extension on the previous study (*Turner et al., 2012*, *2013a*). We only considered swabs from infants in the study, as mothers did not have sufficient sampling resolution relative to their average length of carriage to determine carriage duration. Furthermore, the immune response of mothers to bacterial pathogens is different to children (*Maródi, 2006*), leading to shorter carriage durations (*Gritzfeld et al., 2014*).

To estimate carriage duration from the longitudinal swab data we constructed a set of hidden Markov models (HMMs) with hidden states corresponding to whether a child was carrying a serotype at a given time point, and observed states corresponding to whether a positive swab was observed for this serotype at this time point.

The most general model for the swab data would be a vector with an entry of 0 or 1 for every possible serotype (of 56 observed in the population), corresponding to whether each serotype was observed in the swab at each time point. However, the number of parameters to estimate in this model (with over 6 million states) is much larger than the number of data points (around 14000), and in particular some serotypes have very few positive observations. Instead, we modelled each serotype separately.

The models fitted, and their permitted transitions and emissions are shown in *Figure 2*. In model one, observation $i$ emits state 2 if positively swabbed for the serotype, and state 1 otherwise. The unobserved states correspond to the child 'carrying' and being 'clear' of the serotype respectively. We assume swabs have a specificity of one, so do not show positive culture when the child is clear of the carried serotype; we therefore set the coefficient for the chance of observing positive culture when no bacteria are present to zero ($e_{21} = 0$ in the emission matrix). Model two adds a third state of 'multiple carriage' which is occupied when the serotype and at least one other are being carried. Both models were compared with a version which allows the parameters to covary with whether the child has carried pneumococcus previously. Model three accounts for this explicitly by having separate states and emissions based on whether carriage has previously been observed.

We applied all the models to 19F carriage episodes, as these had the most data available, and calculated the Akaike information criterion (*Akaike, 1974*) for each model that converged. Only the simplest model (model one) converged, as judged by having a positive-definite Hessian and a converged BOBYQA run. The more complex models had lower log-likelihoods: as extensions of the simpler model they should have higher log-likelihoods, so this results was not consistent with model convergence. We tried fitting models two and three using a fixed false positive values slightly greater than zero, this lead to better log-likelihoods, but the models still didn't converge. This failure of the more complex models is probably because most children in the study immediately enter the carrying state, and episodes of dual carriage (when split up by serotype) are rare. Therefore there were not enough events between these carriage states to estimate to the transition and emission intensities, without sensitivity to initial conditions during the fitting.

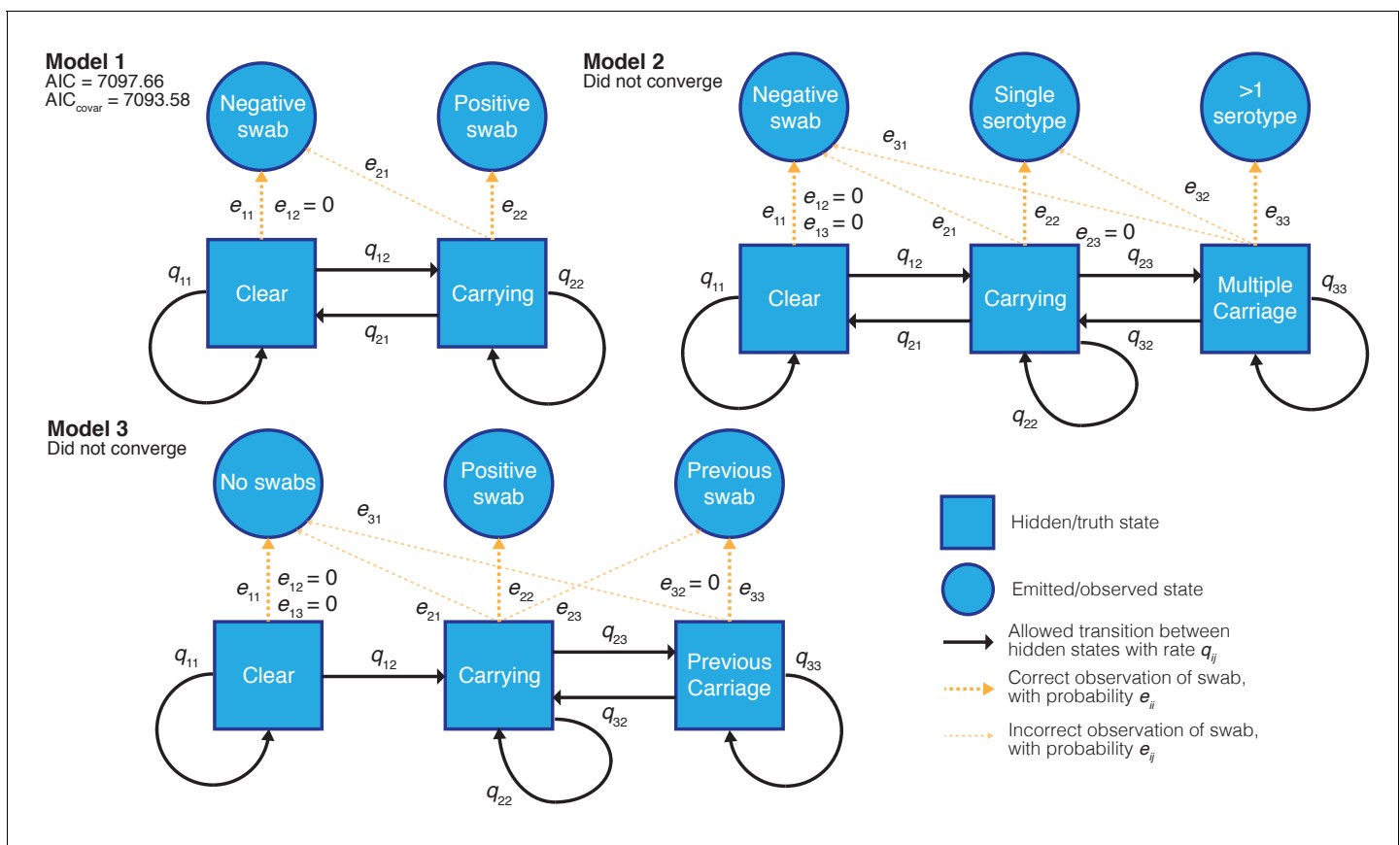

**Figure 2.** Hidden Markov models of swab time series, and their goodness-of-fit. We fitted three different models to the processed time-series data with states, allowed transitions and emissions as shown. We refitted each model allowing the transitions probabilities to covary with the age of the child and whether the child had carried pneumococcus previously. For the converged model the Akaike information criterion (AIC) is shown for the original fit, and when including these covariates (AICcovar).

DOI: https://doi.org/10.7554/eLife.26255.004

We then fitted the best performing model in this test for all serotypes separately. 6A and 6C were treated as a single serotype, as they were not always distinguished in the course of the study. The models for 19F, 23F, 6A/C, 6B, 14 and non-typable (NT) converged, but other serotypes did not have enough observations to successfully fit the parameters of the model. For these less prevalent serotypes we used the transition and emission parameters from the 19F model fitted with the correct observations when reconstructing the most likely route taken through the hidden states. Results were inspected to ensure this did not cause systematic overestimation when compared with previous studies.

We found that the fit for NT swabs produced results which overestimated carriage duration when compared to previously reported estimates. The best fit to the model overestimated the $e_{21}$ parameter, which measures the false negative rate of swabbing, in favour of reduced transition intensities. We therefore fitted the model again, fixing this rate at 0.12. We based this figure on non-typable *Streptococcus pneumoniae* abundance as defined by 16S survey sequencing. At 1% proportional abundance in the sample, 12% came out as culture negative (*Table 1*).

From all the swab data, we estimated that there were a total of 4382 carriage episodes (7.3 per child), of which 2254 had a complete set of AMR data available (*Figure 3*). After removing ten outlier observations from swabs taken accidentally during disease, we were able to match 2157 sequenced genomes with a carriage duration. Duration was positively skewed due to some observations of very long carriage times. We therefore took a monotonic transform of the carriage duration using warped-lmm to maximise the study's power to discover associations and estimate heritability (*Figure 3*). This uses a sum over three nonlinear step functions, plus a linear term, to transform the residuals into Gaussians (*Snelson et al., 2004*).

## Overall heritability of carriage duration is high

The variation in carriage duration $\sigma_P^2$ is partly caused by variance in pneumococcal genetics, and variance in other potentially unknown factors such as host age and host genetics. It is common to write this sum as two components: genetic effects $\sigma_G^2$ and environmental effects $\sigma_E^2$. The proportion of the overall variation which can be explained by the genetics of the bacterium is known as the broad-sense heritability $H^2 = \frac{\sigma_G^2}{\sigma_G^2 + \sigma_E^2}$. Variants which are directly associated with carriage duration independently of other variants (non-epistatic effects) contribute to the narrow-sense heritability $h^2$, which is smaller than the overall broad-sense heritability (*Visscher et al., 2008*).

$H^2$ can be estimated by linear regression on the phenotype of donor-recipient pairs which nearly share their genetics (*Fraser et al., 2014*). However in this dataset we were only able to confidently identify five transmission events, which was not enough to apply this method. Alternatively, analysis of variance of the phenotype between pathogens with similar genetics can be used to estimate heritability (*Anderson et al., 2010*). By applying this to phylogenetically similar bacteria (*Figure 4*), we estimated that $H^2 = 0.634$ (95% CI 0.592–0.686). This implies that the genetics of *S. pneumoniae* is an important factor in determining carriage duration in this population. If environmental conditions are associated with streptococcal genotype between populations (such as host vaccination status) the heritability estimate may differ.

A lower bound on $h^2$ can be calculated by fitting a linear mixed model through maximum likelihood to common SNPs ($h_{\text{SNP}}^2$) (*Lee et al., 2011*; *Manolio et al., 2009*). We used the model in

**Table 1.** Success of culturing unencapsulated *S. pneumoniae*.
Based on having >1% abundance of 16S reads showing the bacteria as being present, 44/361 true positive swabs were not successfully cultured.

| Abundance | Culture positive | Number |
|---|---|---|
| >1% | Cultured | 361 |
| >1% | Not cultured | 44 |
| <1% | Cultured | 56 |
| <1% | Not cultured | 54 |

DOI: https://doi.org/10.7554/eLife.26255.005

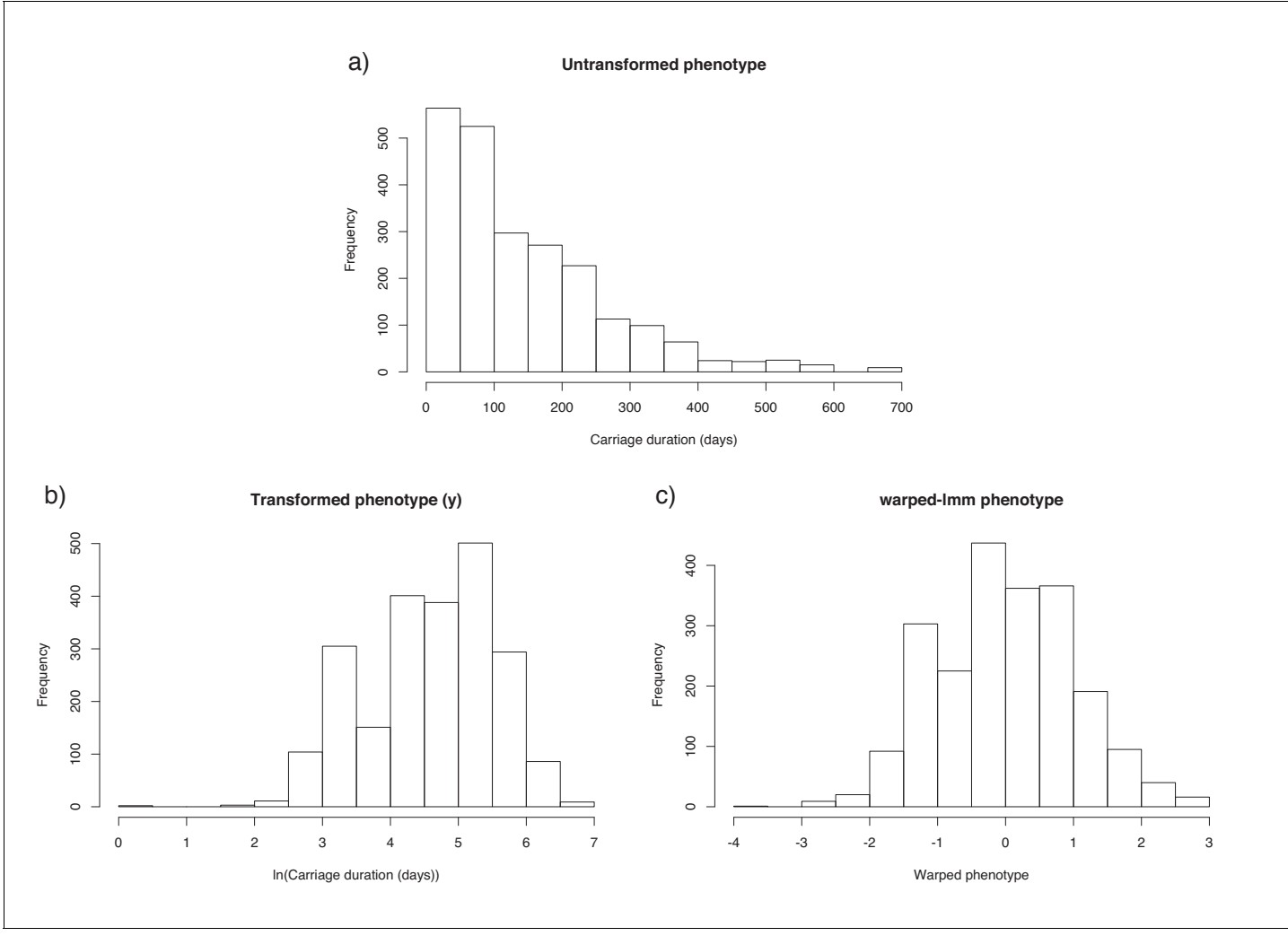

**Figure 3.** Distribution of carriage duration, and effect of monotonic transformation. Panel (**a**) shows a histogram of the inferred carriage duration, (**b**) shows this result after the natural logarithm is taken, and (**c**) after the warping function is applied.

DOI: https://doi.org/10.7554/eLife.26255.006

The following source data and figure supplements are available for figure 3:

**Source data 1.** Sequenced isolates and their untransformed inferred carriage durations.
DOI: https://doi.org/10.7554/eLife.26255.010

**Source data 2.** Sequenced isolates and their warped carriage durations.
DOI: https://doi.org/10.7554/eLife.26255.011

**Figure supplement 1.** Regression diagnostics and outlier removal.
DOI: https://doi.org/10.7554/eLife.26255.007

**Figure supplement 2.** Monotonic warping function from warped-lmm. x-axis shows the centred and normalised input phenotype; y-axis shows corresponding warped value.
DOI: https://doi.org/10.7554/eLife.26255.008

**Figure supplement 3.** Normal quantile-quantile plot of carriage length, and effect of monotonic transformation.
DOI: https://doi.org/10.7554/eLife.26255.009

warped-lmm (*Fusi et al., 2014*) to estimate $h^2_{\mathrm{SNP}}$ for carriage duration data, yielding an estimate of 0.445, consistent with our estimate for $H^2$.

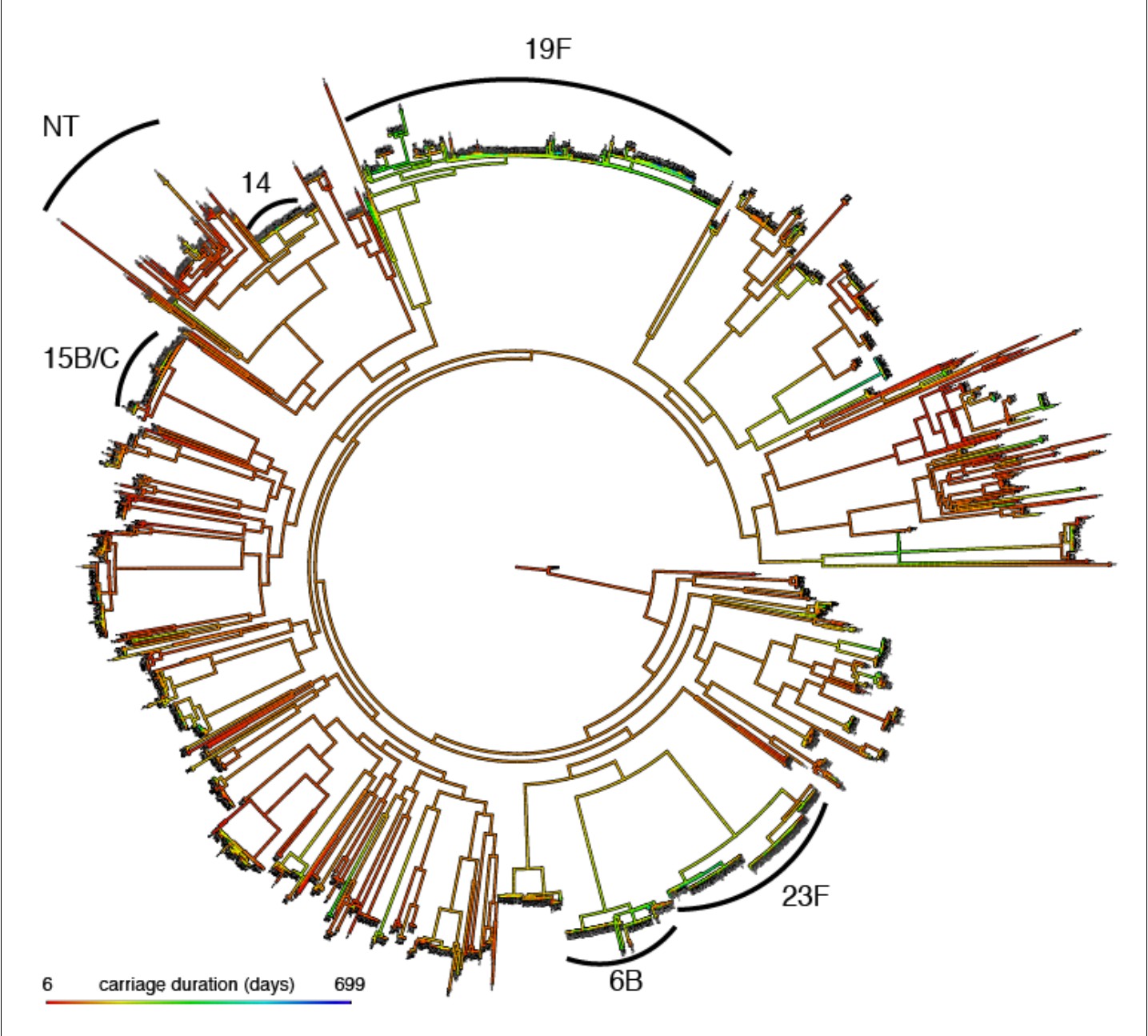

**Figure 4.** Mapping of carriage duration onto phylogeny.  Using the carriage duration as a continuous trait, the ancestral state at every node of the rooted phylogeny was reconstructed. Red branches are carriage for a short time, blue for a long time. Clusters identified in previous analysis have been labelled.

DOI: https://doi.org/10.7554/eLife.26255.012

The following source data and figure supplements are available for figure 4:

**Source data 1.** Phylogenetic tree in Newick format.
DOI: https://doi.org/10.7554/eLife.26255.017
**Figure supplement 1.** Mapping of warped carriage duration onto phylogeny.
DOI: https://doi.org/10.7554/eLife.26255.013
**Figure supplement 2.** Histogram of pairwise patristic distances on the inferred phylogeny.
DOI: https://doi.org/10.7554/eLife.26255.014
**Figure supplement 3.** Change in carriage duration associated with capsule switching events.
DOI: https://doi.org/10.7554/eLife.26255.015
**Figure supplement 4.** Lasso regression plots for lineage effects.
*Figure 4 continued on next page*

*Figure 4 continued*

DOI: https://doi.org/10.7554/eLife.26255.016

## Serotype and drug resistance explain part of the narrow-sense heritability

After calculating the overall heritability, we wished to determine the amount that the specific variation in the pathogen genome contributes to changing carriage duration. In the context of genome wide association studies (GWAS) in bacteria strong linkage-disequilibrium (LD) is present across the entire genome, making it difficult to pinpoint variants associated with carriage duration and not just present in the background of longer or shorter carried lineages (*Chen and Shapiro, 2015*). In *S. pneumoniae*, serotype and antibiogram are correlated with the overall genome sequence (*Brueggemann et al., 2003*; *Chewapreecha et al., 2014a*; *Enright and Spratt, 1998*). If these factors are associated with carriage duration, large sets of variants which define long-carried and short-carried lineages will be correlated with carriage duration in a naive association test (*Chen and Shapiro, 2015*; *Read and Massey, 2014*).

A distinction has therefore been made between variants which evolve convergently and affect a phenotype independently of lineage – termed locus effects – to those which are collinear with a genotype which is associated with the phenotype, termed lineage effects (*Earle et al., 2016*). Locus effects may be associated with a change in carriage duration due to convergent evolution (which may occur through recombination between lineages). In such regions, the causal loci and corresponding phenotypic effects are easier to identify (*Power et al., 2017*). Linear mixed models can be used to find these variants which are associated with a bacterial phenotype independent of lineage; discovery of homoplasic and polygenic variation associated with the phenotype across the entire tree is well powered (*Earle et al., 2016*).

While the high heritability suggests many pathogen variants do affect carriage duration, it does not give information on how many of these will be locus or lineage effects. We mapped carriage duration onto the phylogeny, reconstructing the ancestral state at each node. Consistent with the high heritability of carriage duration we found that carriage length was clearly stratified by lineage (*Figure 4*): we calculated Pagel's lambda as 0.56 ($p<10^{-10}$). We also modelled the evolution of carriage duration along the tree using an Ornstein-Uhlenbeck model, and found that lineage genetics was significantly correlated with the trait (LRT = 952; $p<10^{-10}$)

We first tested for the association of serotype with carriage duration using lasso regression and with a linear-mixed model (LMM). Serotype is correlated with sequence type (*Croucher et al., 2011*) and has previously been associated with differences in carriage duration (*Abdullahi et al., 2012a*; *Turner et al., 2012*). We also included resistance to six antibiotics, the causal element to some of which are known to be associated with specific lineages (*Lees et al., 2016*). These are therefore possible lineage effects which would be unlikely to be found associated under a model which adjusts for population structure (*Chen and Shapiro, 2015*).

Not all serotypes and resistances may have an effect on carriage duration, or there may not be enough carriage episodes observed to reach significance. As including extra predictors in a linear regression always increases the variance explained, we first performed variable selection using lasso regression (*Efron et al., 2004*) to obtain a more reliable estimate of the amount of variation explained. Where a resistance and serotype are correlated and both associated with a change in carriage duration, this will produce a robust selection of the predictors (*Hebiri and Lederer, 2012*).

The selected predictors and their effect on carriage duration are shown in *Table 2*. The total variance explained by these lineage factors was 0.19, 0.178 for serotype alone and 0.092 for resistance alone. When we used genomic partitioning of variance components these were instead estimated to be 0.253, 0.135 and 0.113, respectively. We applied the covariance test (*Lockhart et al., 2014*) to determine which lineage effects were significantly associated with carriage duration and found that 19F, erythromycin resistance, 23F, 6B caused significant ($\alpha<0.05$) increase in carriage duration and being non-typable caused a significant decrease.

Previous studies have used isogenic strains to look for effects of serotype of colonisation and carriage duration independent of genetic background. Resistance to killing (*Weinberger et al., 2009*), growth phenotype (*Hathaway et al., 2012*) and resistance to complement (*Melin et al., 2010*) have

Table 2. Coefficients from lasso regression model of carriage duration.
The mean (intercept) corresponds to a sensitive 6A/C carriage episode, and different serotypes and resistances are perturbations about this mean. Positive effects are expected to have a greater magnitude, due to the positive skew of carriage duration. Rows in bold were significant predictors in the covariance test.

| Factor | Effect on carriage duration (days) |
| --- | --- |
| **Mean (intercept)** | **59.5** |
| **Erythromycin resistance** | **+7.5** |
| Tetracycline resistance | +3.0 |
| Trimethoprim resistance | +2.9 |
| Clindamycin resistance | +1.8 |
| Penicillin intermediate resistance | +1.3 |
| **Serotype 19F** | **+46.9** |
| **Serotype 23F** | **+21.0** |
| **Serotype 6B** | **+16.2** |
| Serotype 14 | +7.2 |
| Serotype 21 | +1.6 |
| Serotype 19B | −0.1 |
| Serotype 18C | −1.9 |
| Serotype 29 | −4.3 |
| Serotype 3 | −4.5 |
| Serotype 4 | −7.2 |
| Serotype 24F | −8.5 |
| Non-typable (NT) | −12.3 |
| Serotype 5 | −18.6 |

DOI: https://doi.org/10.7554/eLife.26255.018

all been shown to affect carriage through serotype rather than genetic background. Conversely, some bacterial genetic variation has been shown to be able to affect colonisation independent of serotype (*Khan et al., 2014*).

We therefore wished to test whether the detected effect of serotype and resistance on carriage duration was entirely mediated through their covariance with lineage, or whether they are independently associated with carriage duration. We first looked for differences in duration over three recent capsule gain/loss events; if there is an effect of serotype independent of genetic background, these would be predicted have the largest difference between serotypes while controlling for the

Table 3. Mean length of carriage, and expected number of carriage episodes within the first two years of life.
Only serotypes with enough data for the HMM fit to converge are shown. Starred observations have a standard error which is larger than the estimated value, indicating low confidence in the estimate.

| Serotype | Sojourn time (days) | Expected number of infections |
| --- | --- | --- |
| **19F** | 292* | 0.85 |
| **23F** | 112 | 0.83 |
| **6A/C** | 76.4 | 0.88 |
| **6B** | 114 | 0.75 |
| **14** | 137* | 0.58 |
| **NT** | 40.6 | 2.05 |

DOI: https://doi.org/10.7554/eLife.26255.019

relatedness of isolates. No significant difference in duration was seen between isolates with or without capsule within the same lineage ($p = 0.39$; *Figure 4*).

However, as these events were limited in number, assumed genetic independence within the clade and occurred only in part of the population, we also performed the same regression as above while also including lineage (defined by discrete population clusters) as a predictor. This therefore allows serotypes which appear in different population clusters to distinguish whether lineage or serotype had a greater effect on carriage duration. The covariance test found that 19F, erythromycin resistance and being non-typable had significant effects on the model (in that order). As these terms enter the model before any lineage specific effect, this suggested these serotypes and resistances are associated with variation in carriage duration independent of background genotype

This lasso-based analysis may be vulnerable to confounding from unmeasured variables which may be associated with the explanatory variables (serotype and resistance). To fully account for the effect of the bacterial genome rather than relying on discrete clusters as covariates in the regression, we performed regression of these lineage effects under an LMM where the relatedness between strains was instead included as a random effect. The predictors had the same order of significance, but only serotype 19F reached genome-wide significance ($p = 3.8 \times 10^{-7}$).

Together, this suggests that the main lineage effect on carriage duration is the serotype, but only some serotypes (19F) have an association independent of genetic background. We also found that erythromycin resistance may be significantly associated with an increased carriage duration. While being a relatively uncommon treatment in this setting (3% of treatments captured), we did not find that other antibiotics were associated. This may be because erythromycin resistance would be expected to cause an almost four order magnitude increase in minimum inhibitory concentration (MIC), whereas other resistance acquisitions have a much smaller effect.

Additionally, we calculated the mean sojourn times (average length of time children are expected to remain in the carrying state of the model with the given serotype) and mean number of carriage episodes from the fit to the HMM for commonly carried serotypes (), which gave results similar to the regression performed above. These estimates are comparable to the previous analysis on a subset of these samples. The majority of carriage episodes were due to five of the seven paediatric serotypes (*Shapiro and Austrian, 1994*), or non-typeable isolates. The results show 19F, 23F and 14 were carried the longest, 6A/C and 6B for intermediate lengths, and NT the shortest.

The overall picture of the first two years of infant carriage is one containing one or two long (over 90 day) carriage episodes of a common serotype (6A/C, 6B, 14, 19F, 23F) and around two short (under a month) carriage episodes of non-typable *S. pneumoniae*. Colonisation by other serotypes seem to cause slightly shorter carriage episodes, though the relative rarity of these events naturally limits the confidence in this inference. That some serotypes are rarer and carried for shorter time periods may be evidence of competitive exclusion (*Hardin, 1960*; *Trzciński et al., 2015*), as fitter serotypes quickly replace less fit serotypes thus leading to reduced carriage duration. The calculated mean carriage duration of NT pneumococci is similar to the minimum resolution we were able to measure by the study design, which suggests carriage episodes may actually be shorter than one month. Unfortunately the only existing study with higher resolution did not check for colonisation by NT pneumococci (*Abdullahi et al., 2012a*).

These estimates are similar to previous longitudinal studies in different populations (*Hill et al., 2010*; *Högberg et al., 2007*; *Melegaro et al., 2007*), though against the Kilifi study our estimates are systematically larger. This may be due to the lower resolution swabbing we performed, or may be because the previous study was unable to resolve multiple carriage (11% of positive swabs). While our heritability estimates are specific to this population due to differences in host, vaccine deployment and transmission dynamics, the similarity of the estimates of serotype effect to those from different study populations suggests our results may be somewhat generalisable.

## Additional loci identified by genome-wide association

To search for locus effects as discussed above, we applied an LMM to all the common SNPs and k-mers in the dataset. The results for SNPs are shown in *Figure 5* and *Table 4*, with 14 loci reaching suggestive significance and two reaching genome-wide significance (top hit $\beta = 0.17$; $p = 2.1 \times 10^{-7}$; MAF = 1%). We also found that 424 k-mers reached genome-wide significance (top hit $\beta = 0.11$; $p = 2.1 \times 10^{-12}$; MAF = 2%), which we filtered to 321 k-mers over 20 bases long to remove low

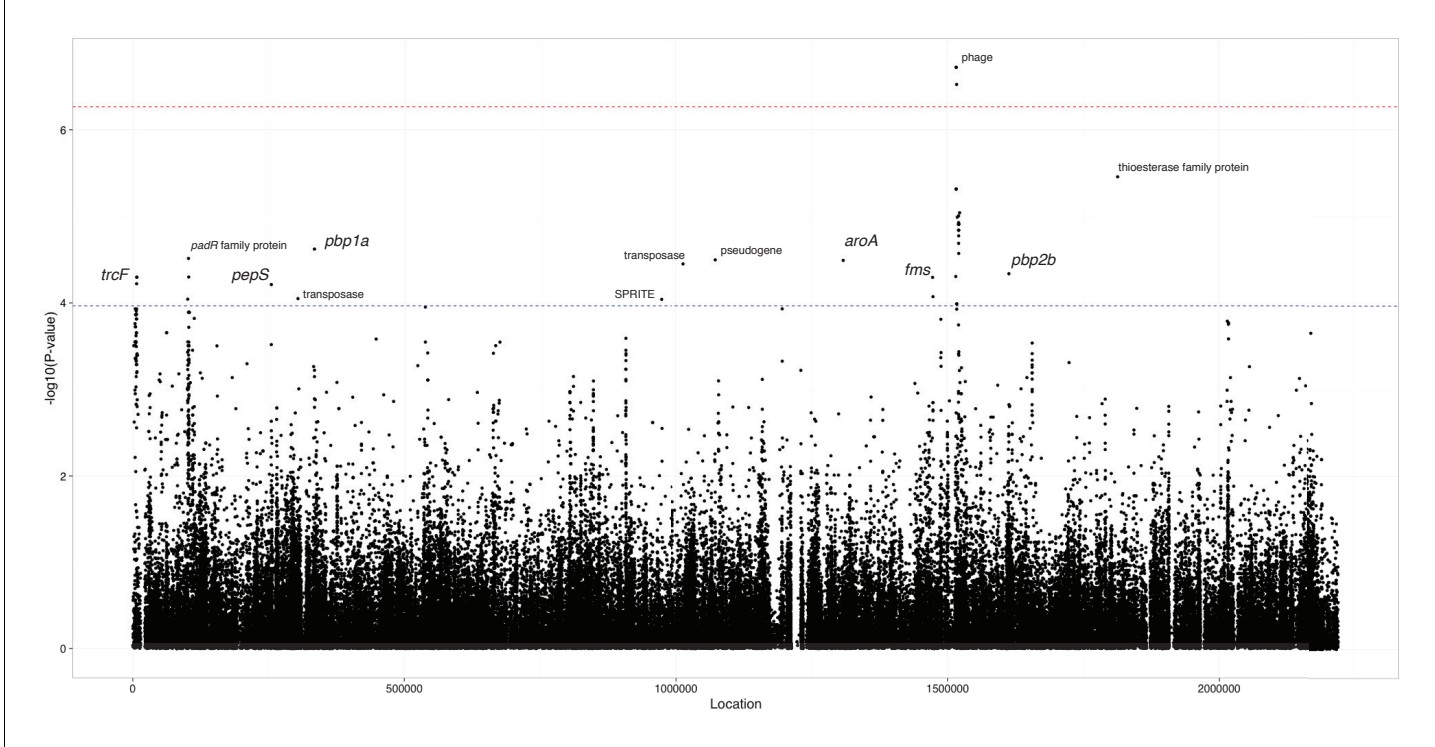

**Figure 5.** Manhattan plot of SNPs associated with carriage duration. The significance of each SNP's association with carriage duration against its position in the ATCC 700669 genome is shown. The red line denotes genome-wide significance ($\alpha<0.05$ Bonferroni corrected with 92487 unique tests), and the blue line suggestive significance (2.3 orders of magnitude below significant, following convention). Loci reaching suggestive significance are labelled with their nearest annotation, as in **Table 4**.

DOI: https://doi.org/10.7554/eLife.26255.020

The following source data and figure supplements are available for figure 5:

**Source data 1.** Plot file for Manhattan plot, with coordinates and -log10 transformed p-values of all tested SNPs.

DOI: https://doi.org/10.7554/eLife.26255.026

**Figure supplement 1.** Possible SNPs associated with lineage and carriage duration.

DOI: https://doi.org/10.7554/eLife.26255.021

**Figure supplement 2.** Distribution of lengths of significant k-mers.

DOI: https://doi.org/10.7554/eLife.26255.022

**Figure supplement 3.** Quantile-quantile plot of association p-values.

DOI: https://doi.org/10.7554/eLife.26255.023

**Figure supplement 4.** Manhattan plots of phage-associated SNPs associated with carriage duration.

DOI: https://doi.org/10.7554/eLife.26255.024

**Figure supplement 5.** Identification of phage in assemblies by blastn hit length.

DOI: https://doi.org/10.7554/eLife.26255.025

specificity sequences (**Figure 5**). To determine their function, we mapped these k-mers to the coordinates of reference sequences.

The only genome-wide significant SNP hits are synonymous changes (MAF = 1%) in the replication module of the prophage in the ATCC 700669 genome (**Croucher et al., 2009**), a highly variable component of the pneumococcal genome (**Croucher et al., 2014a**) (**Figure 5**). The LD structure suggested there were two separate significant signals found in this region. We therefore performed another GWAS conditioning on the top hit to test if there was a second independent signal, but found that the second hit in this region was no longer significant (position 1526024; $p = 2.2 \times 10^{-4}$). The current data is therefore consistent with only a single significant hit to prophage.

The most significant k-mer hits were also located in phage sequence (MAF 2%) and were associated with a reduced duration of carriage. As these mobile genetic elements are less weakly population stratified than other regions of the genome, they are easier to find as locus effects. The LD in

**Table 4.** SNP Locus effects at genome-wide and suggestive significance.
Co-ordinates are with respect to the ATCC 700669 reference genome, and are for the lead SNP in each locus after LD-pruning. Effect sizes are for the warped phenotype.

| Co-ordinate | Nearest annotation | Effect size | P-value | Significance level |
|---|---|---|---|---|
| 6753 | *trcF* | −0.12 | $6.2 \times 10^{-5}$ | Suggestive |
| 254312 | *pepS* | −0.11 | $6.4 \times 10^{-5}$ | Suggestive |
| 303239 | IS630-Spn1 transposase | 0.078 | $9.2 \times 10^{-5}$ | Suggestive |
| 333632 | *pbp1a* | 0.079 | $2.5 \times 10^{-5}$ | Suggestive |
| 971849 | SPRITE repeat region | 0.078 | $9.4 \times 10^{-5}$ | Suggestive |
| 1013978 | IS630-Spn1 transposase | 0.11 | $3.7 \times 10^{-5}$ | Suggestive |
| 1073185 | FM211187.3435 (pseudogene) | 0.086 | $3.3 \times 10^{-5}$ | Suggestive |
| 1308604 | *aroA* | −0.27 | $3.8 \times 10^{-5}$ | Suggestive |
| 1472933 | Upstream of *fms* | −0.23 | $5.3 \times 10^{-5}$ | Suggestive |
| 1473700 | putative glutathione S-transferase | −0.16 | $8.8 \times 10^{-5}$ | Suggestive |
| 1515497 | hypothetical phage protein | −0.099 | $5.2 \times 10^{-5}$ | Suggestive |
| 1516293 | putative phage Holliday junction resolvase | −0.10 | $5.1 \times 10^{-6}$ | Suggestive |
| 1516350 | putative phage Holliday junction resolvase | −0.12 | $2.1 \times 10^{-7}$ | Genome-wide significant |
| 1517063 | phage protein | −0.11 | $3.3 \times 10^{-7}$ | Genome-wide significant |
| 1613197 | *pbp2b* | −0.21 | $4.8 \times 10^{-5}$ | Suggestive |
| 1813192 | thioesterase superfamily protein | −0.12 | $4.8 \times 10^{-6}$ | Suggestive |

DOI: https://doi.org/10.7554/eLife.26255.027

The following source data available for Table 4:

Source data 1. Association results for SNPs, from fast-lmm. DOI: https://doi.org/10.7554/eLife.26255.028

this region is less than in the rest of the genome, as prophage sequence is highly variable within *S. pneumoniae* lineages (*Croucher et al., 2014a*). Multiple independent phage variants may therefore affect carriage duration, which will increase their significance using a LMM. Indeed, the significant results from the LMM (top SNP $p = 2.1 \times 10^{-7}$; top k-mer $p = 2.1 \times 10^{-12}$) are not significant (top SNP $p = 5.1 \times 10^{-6}$; top k-mer $p = 5.7 \times 10^{-8}$) under a model of association using a linear regression with the first 30 principal components as fixed effects to control for population structure rather than random effects, and are strongly associated with the population structure components of the model (highest association $p = 5.2 \times 10^{-75}$ with PC 2).

We postulated that presence of any phage in the genome may cause a reduction in carriage duration. By using the presence of phage as a trait under the same linear mixed model, we however found no evidence of association when correcting for population structure (p=0.35). These results are therefore evidence that infection with a specific phage sequence is associated with a slight decrease in carriage duration. A similar result has previously been found in a genome-wide screen in *Neisseria meningitidis*, where a specific phage sequence was found to affect the virulence and epidemiology of strains (*Bille et al., 2005*; *Bille et al., 2008*). Additionally, previous in vivo tests have shown phage elements to cause a fitness decrease of *S. pneumoniae* during carriage (*DeBardeleben et al., 2014*).

The genetic polymorphisms in the prophage associated with changes in carriage duration, found in 2% of viral sequences, are found within coding sequences inside the phage replication module (*Romero et al., 2009*). It is unlikely the specific variants of these proteins cause a significant difference in cell phenotype, because they are only highly expressed after the prophage is activated, and cell lysis is typically imminent. One explanation for these results is that a subpopulation of prophage do not cause a significant decrease in their host bacterium's carriage duration, which could be due to beneficial 'cargo' genes. Yet previous surveys of pneumococcal prophage have found little

evidence of these elements carrying such sequences (*Croucher et al., 2014a*; *Romero et al., 2009*). One phage protein that has been found to alter the bacterial phenotype is PblB, a phage structural protein that can also mediate bacterial adhesion to human cells (*Loeffler and Fischetti, 2006*). However, *pblB* is within the morphology module (*Romero et al., 2009*) and as an adhesin might be expected to increase carriage duration. Hence the detected association is unlikely to represent expression of viral machinery or cargo genes in the host cell while the prophage is dormant.

Alternatively, the association with only a subset of prophage may be the consequence of sampling. Using a monthly swabbing approach, it was only possible to robustly infer changes in the carriage duration of genotypes that colonise hosts for long periods. Therefore any prophage locus that enhances a virus' ability to infect long carriage duration pneumococci may have an elevated correlation with the variation in the observed phenotype. As phage commonly exhibit high levels of strain specificity (*Duplessis and Moineau, 2001*), this is a plausible mechanism, although the role of the replication module in such host preference is unclear.

An additional mechanism by which prophage can affect host phenotype is by inserting into, and thereby disrupting, functional genes. Pneumococcal prophage frequently insert into *comYC*, thereby preventing the host cell undergoing transformation (*Croucher et al., 2011*; *Croucher et al., 2014b*). Using previous categorisation of the *comYC* gene in this collection into intact versus interrupted or missing (*Croucher et al., 2016*), we found that having an intact *comYC* gene (23% of isolates) was significantly associated with an increased carriage duration using the LMM ($\beta = 0.29$; $p = 1.4 \times 10^{-44}$). The effect size is similar to the associated phage k-mers, but has at a higher allele frequency (hence the increased significance of the result). An interpretation consistent with these findings would be that the effect of phage k-mers is actually through interrupting *comYC*. The k-mers themselves were spread out to lower frequencies due to their sequence variability, and no references used allowed mapping to find the *comYC* interruption directly.

Signals at the suggestive level include *pbp1a* and *pbp2b*, which suggest as above that penicillin resistance may slightly increase carriage duration, but there are not enough samples in this analysis to confirm or refute this. Other signals near genes at a suggestive level included SNPs in *trcF* (transcription coupled DNA repair), *padR* (repressor of phenolic acid stress response), *pepS* (aminopeptidase), *aroA* (aromatic amino acid synthesis), *fms* (peptide deformylase) and a thioesterase superfamily protein. K-mers from erythromycin resistance genes (*ermB*, *mel*, *mef*) were expected to reach significance from the above analysis, but did not: it has however previously been shown that the power to detect these elements in a larger sample set taken from the same population is limited due to the multiple resistance mechanisms and stratification of resistance with lineage (*Lees et al., 2016*).

The test statistic from fast-lmm roughly followed the null-hypothesis, with the exception of the significant phage k-mers (*Figure 5*). However there is limited power to detect effects associated with both the lineage and phenotype. This effect has been previously noted, and while LMMs have improved power for detecting locus specific effects they lose power when detecting associated variants which segregate with background genotype (*Earle et al., 2016*).

To search for candidate regions which may be independently associated with both a lineage and increased carriage duration, we ran an association test using a set number of fixed effects as the population structure correction. This is expected to have higher power than an LMM for true associated variants on ancestral branches, but will also increase the number of false positives (variants co-occurring on these branches which do not directly affect the carriage duration themselves). We also tested SNPs for their association with those principal components which were themselves significantly associated with carriage duration, and therefore may be driving the lineage associations (*Earle et al., 2016*).

The most highly associated SNPs were in all three *pbp* regions associated with $\beta$-lactam resistance, the capsule locus, *recA* (DNA repair and homologous recombination), *bgaA* (beta-galactosidase), *phoH*-like protein (phosphate starvation-inducible protein), *ftsZ* (cell division protein) and *groEL* (chaperonin). As 19F, the serotype most associated with carriage duration, is predominantly the $\beta$-lactam resistant PMEN14 lineage the *pbp* association may be driven through strong LD between with this serotype. *Figure 5—figure supplement 1* shows the analysis of SNPs which may be driving significant lineage associations – this also suggested *dnaB* (DNA replication) may be associated with altered carriage duration. Associated k-mers were also found in *phtD* (host cell surface adhesion), *mraY* (cell wall biosynthesis), *tlyA* (rRNA methylase), *zinT* (zinc recruitment), *adcA* (zinc

recruitment) and *recJ* (DNA repair). Additionally we found k-mers in the bacteriocin *blpZ* and immunity protein *pncM* (**Bogaardt et al., 2015**) to be associated with variability in carriage duration. This could be evidence that intra-strain competition occurs within host via this mechanism, consistent with previous in vitro mouse models (**Dawid et al., 2007**).

It is not possible to determine whether variation in these genes is associated with a change in carriage duration or if the variation is present in longer carried, generally more prevalent lineages. For example, *β*-lactam resistance may appear associated as the long carried lineages 19F and 23F are more frequently resistant, or it may genuinely provide an advantage in the nasopharynx that extends carriage duration independent of other factors. Future studies of carriage duration, or further experimental evidence will be needed to determine which is the case for these regions.

Antigenic variation in known regions (of *pspA*, *pspC*, *zmpA* or *zmpB*) may be expected to cause a change in carriage duration (**Lipsitch and O'Hagan, 2007**), however we found none of these to be associated with a change in carriage duration. This was likely due to stratification of variation in these regions with lineage, but may also be caused by a larger diversity of k-mers in the region reducing power to detect an association.

## Child age independently affects variance in carriage duration

Finally, we wished to determine the importance of two environmental factors which are known to contribute to variance in this phenotype: child age and whether the carriage episode is the first the child has been exposed to (**Abdullahi et al., 2012a**; **Abdullahi et al., 2012b**; **Turner et al., 2012**). These have been applied throughout the analysis as covariates, both in the estimation of carriage episodes and in associating genetic variation with change in carriage duration.

We applied linear regression to these factors while using the first 30 PCs to correct for the effect of the bacterial genome, which showed they were both significantly associated with carriage duration as expected (age $p = 3.9 \times 10^{-7}$; previous carriage $p = 2.5 \times 10^{-8}$). Using the linear mixed model to control for bacterial genotype both factors were again significant (LRT = 26.4; $p = 1.8 \times 10^{-6}$). Together, they explained 0.046 of variation in carriage duration. As found previously, increasing child age contributes to a decrease in the duration of carriage episodes. From a mean of 68 days long, we calculated a drop of 19 days after a year, and 32 days after two years. Extrapolating, this causes carriage episodes longer than two days to cease by age 11 (**Figure 6**). Previous carriage of any serotype was estimated to cause an increase in the duration of future carriage episodes, though previous studies have found no overall effect (**Weinberger et al., 2008**). It has previously been shown that prior exposure to non-typables in this cohort make colonisation by another non-typable occur later, and for a shorter time (**Turner et al., 2012**). The positive effect observed in this analysis is therefore likely to be an artefact due to subsequent carriage episodes being more likely to be due to typable pneumococci.

Additional environmental factors that explain some of the remainder of the variance may include the variation of the host immune response and interaction with other infections or co-colonisation. In particular, co-infection influenza A was not recorded but is known to affect population dynamics within the nasopharynx (**Kono et al., 2016**). Fundamentally, imprecise inference of the carriage duration will limit our ability to fully explain its variance.

## Discussion

Other than serotype, the genetic determinants of pneumococcal carriage duration were previously unknown. By developing models for longitudinal swab data and combining the results with whole genome sequence data we have quantified and mapped the genetic contribution to the carriage duration of *S. pneumoniae*. We found that despite a range of other factors such as host age which are known to cause carriage duration to differ, sequence variation of the pneumococcal genome explains most of this variability (63%). Common serotypes and resistance to erythromycin cause some of this effect (19% total), as does the presence or absence of particular prophage sequence in the genome. *Table 5* summarise the sources we found to be significantly associated with variation in carriage duration.

We provide a quantitative estimate of how closely transmission pairs share their carriage duration, and show evidence for differences both between and within serotypes. The implication of phage as having a significant effect on carriage duration has interesting corollaries on pneumococcal genome

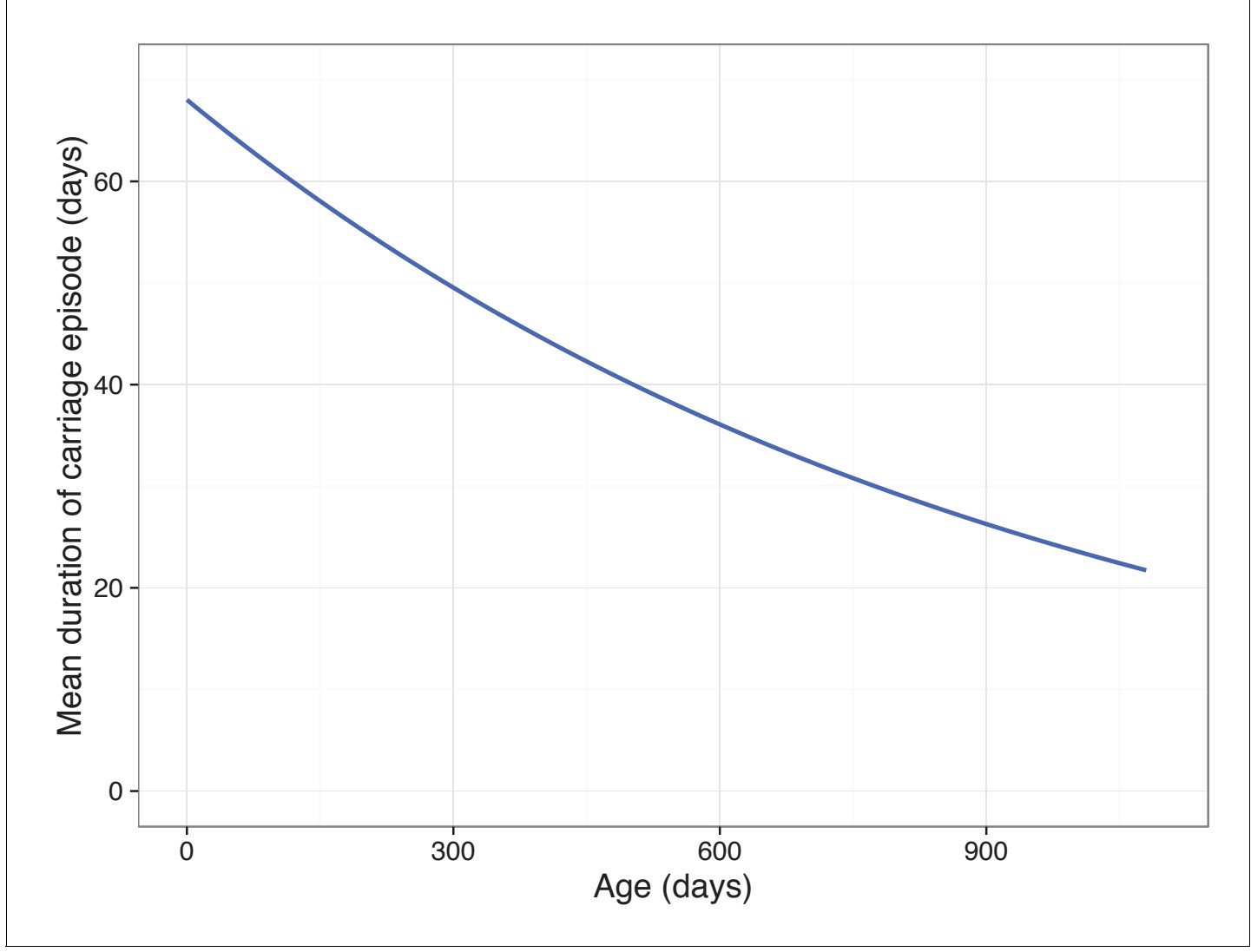

**Figure 6.** Predicted mean carriage duration as a function of child age. Fit is an exponential decay over the first two years of life, using the decay rate inferred from a linear regression of log(carriage duration).

DOI: https://doi.org/10.7554/eLife.26255.029

diversification through frequent infection and loss of prophage, even during carriage episodes in this dataset.

Investigating a mechanism for the prophage association, we found that having an intact *comYC* gene, which is frequently interrupted by prophage causing loss of function of the competence system, was associated with increased carriage duration. While the competence system is observed to remain intact over the evolutionary history of the species, these disruptive mutations spread irreversibly through the population as competent bacteria can acquire the mutation, and non-competent bacteria can no longer reverse it through recombination (*Croucher et al., 2014b*). Selection must therefore maintain the function at this locus over short timescales, and an increased carriage duration may be evidence of this. We therefore hypothesise that the associated prophage sequences may affect carriage duration through disruption of the competence system, without which deleterious mutations will accumulate in the population due to Muller's ratchet.

The results presented here have important implications for the modelling of pneumococcal transmission and their response to perturbation of the population by vaccine. Importantly, our analysis of heritability shows that variants other than serotype affect carriage duration, consistent with recent theoretical work (*Lehtinen et al., 2017*). Here we have shown that these alleles do exist in a natural

**Table 5.** Summary of variance of carriage duration explained by genetic and environmental factors.

$H^2$ encompasses all rows, other than the measured environmental effects. For each variant component the method used to estimate it is reported: CPP - closest phylogenetic pairs; LMM - variance component using a linear mixed model with pathogen genotype as random effects; $R^2$ - linear regression using lasso to select predictors.

| Source | Of which is | Total variance explained | Proportion of total heritability explained |
|---|---|---|---|
| Total heritability ($H^2$) | | 0.634 (CPP) | 1.00 |
| | Common SNP heritability ($h^2_{SNP}$) | 0.438 (LMM) | 0.691 |
| | Serotype and resistance | 0.190 ($R^2$)/0.253 (LMM) | 0.300 ($R^2$)/0.399 (LMM) |
| | Serotype only | 0.178 ($R^2$)/0.135 (LMM) | 0.281 ($R^2$)/0.213 (LMM) |
| | Resistance only | 0.092 ($R^2$)/0.113 (LMM) | 0.145 ($R^2$)/0.178 (LMM) |
| | Phage k-mers | 0.067 (LMM) | 0.106 |
| | Intact *comYC* | 0.127 (LMM) | 0.201 |
| Measured environmental effects | Age and previous carriage | 0.046 ($R^2$) | - |

DOI: https://doi.org/10.7554/eLife.26255.030

population, and also identified candidates for the loci which fulfil this role. Together these studies suggest that variants exist in the pneumococcal genome which alter carriage duration, which in turn is linked to antibiotic resistance.

We were not able to fully explain the basis for heritability of carriage duration for a number of reasons. The close association of the phenotype with lineage limited our power to fine-map lineage associated variants other than capsule type which may affect carriage duration. Meta-analysis with more large studies with higher resolution may help to resolve these issues: we are conducting a similar study in Cape Town, South Africa which will combine sequence data with two-weekly swabs and will be compared to these results in future. Additional environmental factors that explain some of the remainder of the variance may include the variation of the host immune response and interaction with other infections or co-colonisation. In particular, co-infection with influenza A was not recorded but is known to affect population dynamics within the nasopharynx (*Kono et al., 2016*).

This is a phenotype which would have been difficult to assay by traditional methods such as in an animal model due to the cohort size needed and the length of time experiments would need to be run for. By instead using genome-wide association study methods we have been able to quantitatively investigate a complex phenotype in a natural population. We believe that the analysis of heritability and variance explained in a phenotype of interest, as presented here, will be an important part analysis of complex bacterial traits in future studies.

# Materials and methods

## Sample collection

The study population was a subset of infants from the Maela longitudinal birth cohort (*Turner et al., 2013a*), and was split into two cohorts. In the 'routine' cohort, 364 infants were swabbed monthly from birth, 24 times in total. All swabs were cultured and serotyped using the latex sweep method (*Turner et al., 2013b*). In the 'immunology' cohort 234 infants were swabbed on the same time schedule, but cultured and serotyped following the World Health Organisation (WHO) method (*Turner et al., 2012*). Non-typable pneumococci were confirmed by bile solubility, optochin susceptibility and Omniserum Quellung negative. For both cohorts phenotypic drug resistance to six antibiotics was available (chloramphenicol, $\beta$-lactams, clindamycin, erythromycin, trimethoprim and tetracycline). 3161 randomly selected pneumococcal positive swabs from the study population have been previously sequenced, 2175 of which were from these longitudinal infant samples (*Chewapreecha et al., 2014a*).

### Converting swab data into a time series

Latex sweeps could not differentiate 6A and 6C serotypes, so we treated these as a single serotype when detected by this method (in WHO serotyping PCR was used to differentiate these serotypes). 15B and 15C serotypes spontaneously interconvert, so were combined. We removed two duplicated swabs (08B09098 from the immunology cohort; 09B02164 from the routine observation cohort).

To get a good fit of the HMM, we normalised observation times for each sample. Defining infant birth as $t = 0$, subsequent sampling times $t_i$ were measured in days, and normalised to have a variance of one. The actual (untransformed) carriage duration in days was used as initial phenotype $y$.

### Hidden markov model of time series

We modelled the time series of swab data using a continuous-time HMM, as implemented in the R package msm (RRID:SCR_015500) (*Jackson, 2011*). Unobserved (true) states correspond to whether the child is carrying bacteria in their nasopharynx, and observed (emitted) states correspond to whether a positive swab was seen at each point. Transition probabilities between each state **Q** and the emission probabilities **E** are jointly estimated by maximum likelihood using the BOBYQA algorithm. We then constructed the most likely path through the unobserved states for each child using the Viterbi algorithm (*Forney, 1973*) with the observed data and estimated model parameters. Assuming that continuous occupation of the carried state corresponded to a single carriage episode, we calculated the duration for each such episode from the inferred true states.

### Processing genetic data

For each isolate with an inferred carriage duration (N = 2175) we extracted SNPs from the previously generated alignment against the ATCC 700669 genome (*Chewapreecha et al., 2014b*). Consequences of SNPs were annotated with VEP (RRID:SCR_007931), using a manually prepared reference (*McLaren et al., 2010*). A phylogenetic tree was generated from this alignment using FastTree (RRID:SCR_015501) under the GTR +gamma model (*Price et al., 2009*). The carriage duration was mapped on to this phylogeny using phytools (RRID:SCR_015502) (*Revell, 2013*). We then filtered the sites in the alignment to remove any where the major allele was an N, any sites with a minor allele frequency lower than 1%, and any sites where over 5% of calls were missing. This left 115210 sites for association testing and narrow-sense heritability estimation.

We counted 68M non-redundant k-mers with lengths 9–100 from the de novo assemblies of the genomes using a distributed string mining algorithm (*Seth et al., 2014*; *Välimäki and Puglisi, 2012*). We filtered out low frequency variants removing any k-mers with a minor allele frequency below 2%, leaving 17M for association testing.

We identified the presence of phage by performing a blastn of the de novo assemblies against a reference database of phage sequence (*Croucher et al., 2016*). If the length of the top hit was over 5000 we defined the isolate as having phage present (*Figure 5*).

### Transformation of carriage duration phenotype

As we aimed to fit a multiple linear regression model to the carriage duration $y$ at each genetic locus $k$, we first ensured the data was appropriate for this model. The phenotype distribution was positively skewed, with an approximately exponential distribution (*Figure 3*). Residuals were therefore non-normally distributed, potentially reducing power (*McCulloch, 2003*). In the regression setting, a monotonic function can be applied to transform the response variable to avoid this problem. We took the natural logarithm of the carriage duration

$$\hat{\mathbf{y}} = \ln(\mathbf{y})$$

which led to the residuals being much closer to being normally distributed (*Figure 3*). We applied the same transformation to child age, when it was used as a covariate in association.

### Estimation of heritability

We estimated broad sense heritability $H^2$ with the ANOVA-CPP method in the patherit R package (*Mitov and Stadler, 2016*), using a patristic distance cutoff of 0.04 (*Figure 4*). To test the effect of lineage genetics we used the patherit package to fit an Ornstein-Uhlenbeck model of the warped

carriage duration along the phylogeny. We compared the likelihood of the full fit to that with no genetic effect on the trait ($\sigma_G^2 = 0$) using a likelihood ratio test (LRT) with one degree of freedom.

To estimate the SNP-based heritability $h_{\mathrm{SNP}}^2$ we applied a linear mixed model, which uses the genomic relatedness matrix (as calculated from SNPs passing filtering) as random effects. We used the implementation in warped-lmm (RRID:SCR_015503) (*Fusi et al., 2014*), which learns a monotonic transform as it fits the model to the data to ensure residuals are normally distributed (*Figure 3*). We therefore used the untransformed phenotype $y$ as the input. Child age and whether previous carriage had occurred were included as covariates. We also estimated $h_{\mathrm{SNP}}^2$ using LDAK (RRID:SCR_015504) (*Speed et al., 2012*) with default settings, which gave an estimate of 0.437 (<1% difference from the warped-lmm estimate).

## Association of antimicrobial resistance and serotype with carriage duration

We encoded all 56 observed serotypes (including non-typables) and resistance to the six antibiotics as dummy variables. We used 6A/C as the reference level, as this had a mean carriage duration close to the grand mean in previous analysis. Orthogonal polynomial coding was used for the latter four antibiotics, where resistance could be intermediate or full. We then regressed this design matrix **X** was against the transformed carriage duration $\hat{y}$. We removed three observations with low carriage lengths due to a delayed initial swab, and seven observations with leverages of one (*Figure 3*).

We performed variable selection using lasso regression (*Efron et al., 2004*), implemented in the R package glmnet (RRID:SCR_015505) (*Friedman et al., 2010*). We used leave-one-out cross-validation to choose a value for the $\ell_1$ penalty; the value one standard error above the minimum cross-validated error (*Tibshirani et al., 2001*) was selected ($\lambda = 0.033$; *Figure 4*). The 20 predictors with non-zero coefficients in the model at this value of $\lambda$ (*Table 2*) were used in a linear regression to calculate the multiple $R^2$, which corresponds to the proportion of variance explained by these predictors.

To estimate the variance components from serotype and resistance we used genomic partitioning (*Yang et al., 2011*), as implemented in LDAK. We used SNPs in the capsule locus to calculate a kinship matrix approximating the contribution from serotype variation. For antibiotic resistance we used SNPs in the *pbp* genes, *dyr* gene and ICE transposon to calculate a kinship matrix. Restricted maximum likelihood was used to estimate the variance explained by each of these components.

Capsule switch events had been previously identified by first reconstructing of the ancestral state of the serotype at each node through maximum parsimony (*Chewapreecha et al., 2014a*). For each node involving loss or gain of the capsule, those with at least one child being a tip were selected to find recent switches (all were capsule gain). The carriage duration of all unencapsulated children of the identified node were used as the null distribution to calculate an empirical p-value for the switched isolate. P-values were combined using Fisher's method (*Rosenthal, 1978*).

## Genome wide association of carriage duration

We used the linear mixed model implemented in fast-lmm (RRID:SCR_015506) (*Lippert et al., 2011*) to associate genetic elements with carriage duration, independent of overall lineage effects. We used the warped phenotype as the response, the kinship matrix (calculated from SNPs) as random effects, and variant presence, child age and previous carriage as fixed effects. For SNPs we used a Bonferroni correction with $\alpha < 0.05$ and an N of 92487 phylogenetically independent sites to derive a genome-wide significance cutoff of $p < 5.4 \times 10^{-7}$, and a suggestive significance cutoff (*Lander and Kruglyak, 1995*; *Stranger et al., 2011*) of $p = 1.1 \times 10^{-4}$. We tested pairwise LD between the significant SNPs by calculating the $R^2$ between them. We removed those with $R^2 > 0.2$, assuming these represented the same underlying signal, to define the significant loci. To perform conditional analysis we used the pattern of the most significant SNP as a fixed-effect, removed it from the test and kinship estimation, and re-ran the mixed model on all other sites.

For k-mers we counted 5254876 phylogenetically independent sites, giving a genome wide significance cutoff of $9.5 \times 10^{-9}$. We used blastn (RRID:SCR_001598) with default settings to map the significant k-mers to seven reference genomes (ATCC 700669, INV104B, OXC141, SPNA45, Taiwan19F, TIGR4 and NT_110_58), and the possible Tn*916* sequences (*Croucher et al., 2011*).

To search for variants with some level of lineage independence we used SEER (RRID:SCR_015499) (*Lees et al., 2016*). To correct for population structure we used the patristic distances from

the phylogenetic tree as the kinship matrix, which we then projected into 30 dimensions using metric multidimensional scaling. The coordinates of the samples in this space were used as covariates in SEER's linear regression. We performed association tests on SNPs and k-mers with MAF > 1% using multiple linear regression, and report the top hits with $p<10^{-14}$. Significant k-mers were mapped as above.

## Code and data availability

All code used for analysis is available on github, along with inferred carriage duration for each sample (*Lees, 2017*). A copy is archived at https://github.com/elifesciences-publications/carriage-duration.

## Acknowledgements

We would like to thank Susannah Salter for sharing data on non-typable culture positive rates. We also wish to thank Ben Cooper, Doug Speed, and the attendees of the 8th Permafrost workshop (particularly Jukka Corander, Christophe Fraser, Sonja Lehtinen and Johan Pensar) for comments on this work. Work at the Wellcome Trust Sanger Institute was supported by Wellcome Trust (098051). JAL was supported by a Medical Research Council studentship grant (1365620). NJC was supported by a Sir Henry Dale Fellowship, jointly funded by the Wellcome Trust and the Royal Society (grant number 104169/Z/14/Z). PT was supported by the Wellcome Trust (Grant No. 083735/Z/07/Z). SMRU is part of the Mahidol Oxford Tropical Medicine Research Unit supported by the Wellcome Trust.

## Additional information

### Funding

| Funder | Grant reference number | Author |
|---|---|---|
| Wellcome Trust | 098051 | John A Lees<br>Julian Parkhill<br>Stephen D Bentley |
| Medical Research Council | 1365620 | John A Lees |
| Royal Society | 104169/Z/14/Z | Nicholas J Croucher |
| Wellcome Trust | 104169/Z/14/Z | Nicholas J Croucher |
| Wellcome Trust | 083735/Z/07/Z | Paul Turner |

The funders had no role in study design, data collection and interpretation, or the decision to submit the work for publication.

### Author contributions

John A Lees, Conceptualization, Data curation, Formal analysis, Methodology, Writing—original draft, Writing—review and editing; Nicholas J Croucher, Investigation, Writing—review and editing; David Goldblatt, Resources, Writing—review and editing; François Nosten, Julian Parkhill, Resources, Supervision, Writing—review and editing; Claudia Turner, Conceptualization, Resources, Project administration, Writing—review and editing; Paul Turner, Conceptualization, Data curation, Supervision, Project administration, Writing—review and editing; Stephen D Bentley, Conceptualization, Supervision, Writing—review and editing

### Author ORCIDs

John A Lees, http://orcid.org/0000-0001-5360-1254
Nicholas J Croucher, http://orcid.org/0000-0001-6303-8768
François Nosten, http://orcid.org/0000-0002-7951-0745
Paul Turner, http://orcid.org/0000-0002-1013-7815

## Ethics

**Human subjects:** Written informed consent was obtained from the mothers prior to study enrolment. Ethical approval was granted by the ethics committees of the Faculty of Tropical Medicine, Mahidol University, Thailand (MUTM-2009-306) and Oxford University, UK (OXTREC-031-06).

## Decision letter and Author response

Decision letter https://doi.org/10.7554/eLife.26255.043
Author response https://doi.org/10.7554/eLife.26255.044

# Additional files

## Major datasets

The following previously published datasets were used:

| Author(s) | Year | Dataset title | Dataset URL | Database, license, and accessibility information |
|---|---|---|---|---|
| Chewapreecha C, Harris SR, Croucher NJ, Turner C, Marttinen P, Cheng L, Pessia A, Aanensen DM, Mather AE, Page AJ, Salter SJ, Harris D, Nosten F, Goldblatt D, Corander J, Parkhill J, Turner P, Bentley SD | 2014 | Dense genomic sampling identifies highways of pneumococcal recombination | https://www.ncbi.nlm.nih.gov/sra/?term=ERP000435 | Publicly available in the NCBI Sequencing Read Archive (SRA) (accession no. ERP000435) |
| Chewapreecha C, Harris SR, Croucher NJ, Turner C, Marttinen P, Cheng L, Pessia A, Aanensen DM, Mather AE, Page AJ, Salter SJ, Harris D, Nosten F, Goldblatt D, Corander J, Parkhill J, Turner P, Bentley SD | 2014 | Dense genomic sampling identifies highways of pneumococcal recombination | https://www.ncbi.nlm.nih.gov/sra/?term=ERP000483 | Publicly available in the NCBI Sequencing Read Archive (SRA) (accession no. ERP000483) |
| Chewapreecha C, Harris SR, Croucher NJ, Turner C, Marttinen P, Cheng L, Pessia A, Aanensen DM, Mather AE, Page AJ, Salter SJ, Harris D, Nosten F, Goldblatt D, Corander J, Parkhill J, Turner P, Bentley SD | 2014 | Dense genomic sampling identifies highways of pneumococcal recombination | https://www.ncbi.nlm.nih.gov/sra/?term=ERP000485 | Publicly available in the NCBI Sequencing Read Archive (SRA) (accession no. ERP000485) |

| | | | | |
|---|---|---|---|---|
| Chewapreecha C, Harris SR, Croucher NJ, Turner C, Marttinen P, Cheng L, Pessia A, Aanensen DM, Mather AE, Page AJ, Salter SJ, Harris D, Nosten F, Goldblatt D, Corander J, Parkhill J, Turner P, Bentley SD | 2014 | Dense genomic sampling identifies highways of pneumococcal recombination | https://www.ncbi.nlm.nih.gov/sra/?term=ERP000487 | Publicly available in the NCBI Sequencing Read Archive (SRA) (accession no. ERP000487) |
| Chewapreecha C, Harris SR, Croucher NJ, Turner C, Marttinen P, Cheng L, Pessia A, Aanensen DM, Mather AE, Page AJ, Salter SJ, Harris D, Nosten F, Goldblatt D, Corander J, Parkhill J, Turner P, Bentley SD | 2014 | Dense genomic sampling identifies highways of pneumococcal recombination | https://www.ncbi.nlm.nih.gov/sra/?term=ERP000598 | Publicly available in the NCBI Sequencing Read Archive (SRA) (accession no. ERP000598) |
| Chewapreecha C, Harris SR, Croucher NJ, Turner C, Marttinen P, Cheng L, Pessia A, Aanensen DM, Mather AE, Page AJ, Salter SJ, Harris D, Nosten F, Goldblatt D, Corander J, Parkhill J, Turner P, Bentley SD | 2014 | Dense genomic sampling identifies highways of pneumococcal recombination | https://www.ncbi.nlm.nih.gov/sra/?term=ERP000599 | Publicly available in the NCBI Sequencing Read Archive (SRA) (accession no. ERP000599) |

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
