## [Decision Letter]

Thank you for submitting your article "Genome-wide identification of lineage and locus specific variation associated with pneumococcal carriage duration" for consideration by *eLife*. Your article has been reviewed by three peer reviewers, and the evaluation has been overseen by a Reviewing Editor and Wendy Garrett as the Senior Editor. The following individual involved in review of your submission has agreed to reveal his identity: Daniel Wilson.

The reviewers have discussed the reviews with one another and the Reviewing Editor has drafted this decision to help you prepare a revised submission.

The reviewers agree that this is an interesting and novel analysis of an impressive data set. The study sheds light on an important problem in clinical microbiology, the genetic determinants of carriage duration in pneumococcus, and the work should thus appeal to modelers in bacterial genomics and epidemiology. The reviewers find the conclusions fairly convincing and note that the association between carriage duration and the presence of prophage is an especially intriguing result that could inspire future investigations.

The reviewers note, however, the statistical challenges inherent in this type of study. They provide several suggestions for how the statistics could be improved or clarified to carefully delineate the support for different conclusions. An overriding concern is that the limitations of the statistics, and especially the inability of associations to demonstrate causality, be accurately communicated. The reviewers also recommend discussing the potential biology of the phage association in more depth.

Essential revisions and major points:

1) Exploring the inclusion of some components (e.g. serotype, resistance, lineage), but not all components of genetic variation (i.e. the rest of the genome) on phenotype using a Lasso is fairly ad hoc, making it difficult to have confidence in the conclusions. Besides the uncertainty in distinguishing the effects of these loci from others not included, there is also the problem that unmeasured, heritable confounders may influence results, because population stratification may not be adequately controlled. Lasso can make arbitrary choices between approximately equally good variables even if all loci were included, making it good for predicting phenotype, but bad for identifying candidate causal loci. It would be appropriate to mention that in the Lasso analysis, there is less robustness to potential confounding with unmeasured variables that may be associated with the significant regressors. Inferences of causality (e.g. subsection “Serotype and drug resistance explain part of the narrow-sense heritability”, seventh paragraph) are therefore likely to be overstated. For more robust inference, I would suggest testing the significance of these explanatory variables over and above what is explained by, e.g. the top 30 PCs or (better) in a lineage mixed model. If these analyses do not support the same conclusions, that is important to highlight.

2) The authors need to be more careful/speculative in their description of their findings, especially in the Abstract. For example, with respect to the role of host factors, this was far from exhaustive in this study, and there are many more host features, that were the data available could have accounted for more of the phenotype variation, so I believe it is more accurate to state "We estimated that pneumococcal genomic variation accounted for 63% of the phenotype variation, whereas *the* host traits *considered here* accounted for less than 5%." (Clearly, the inclusion of epidemiological metadata, if any are available, would enrich the story.)

The second statement in the Abstract that is overstated is 'A pan-genome-wide association study identified prophage sequences as significantly decreasing carriage duration independent of serotype.' Whereas without any evidence of causation, I believe they can only state that they have "identified prophage sequences that significantly associated with decreasing carriage duration independent of serotype." The same idea holds for the association of erithromycin resistance and carriage – the causality is not straightforward. Under the theory of Lehtinen et al. (2017), long durations drive resistance due to the sensitivity of strains with long carriage durations to the fitness effects of antibiotic use, not the other way around.

3) The association of the polymorphic phage with carriage is intriguing and deserves more treatment. Phages likely affect the strain in very different ways, some potentially extending carriage duration with other others decreasing it. Can phages in the database be partitioned into categories based on predicted phenotypic impact and the association retested? What was the genetic polymorphism in the phage genes that associated with carriage duration? How many strains was this in, i.e., was it a large effect in a small number of samples or a small affect in a large number? What was the mean carriage duration for strains with and without it (similar to that shown for serotype in Table 2). Phage often encode specific virulence or immune evasion factors, does this phage carry these? (Ideally a lab-based assay could be performed on the clinical strains to provide some functional information about its potential effect, or a second collection of isolates could be used to validate the finding, but neither is required.)

---

## [Author Response]

Essential revisions and major points:

1) Exploring the inclusion of some components (e.g. serotype, resistance, lineage), but not all components of genetic variation (i.e. the rest of the genome) on phenotype using a Lasso is fairly ad hoc, making it difficult to have confidence in the conclusions. Besides the uncertainty in distinguishing the effects of these loci from others not included, there is also the problem that unmeasured, heritable confounders may influence results, because population stratification may not be adequately controlled. Lasso can make arbitrary choices between approximately equally good variables even if all loci were included, making it good for predicting phenotype, but bad for identifying candidate causal loci. It would be appropriate to mention that in the Lasso analysis, there is less robustness to potential confounding with unmeasured variables that may be associated with the significant regressors. Inferences of causality (e.g. subsection “Serotype and drug resistance explain part of the narrow-sense heritability”, seventh paragraph) are therefore likely to be overstated. For more robust inference, I would suggest testing the significance of these explanatory variables over and above what is explained by, e.g. the top 30 PCs or (better) in a lineage mixed model. If these analyses do not support the same conclusions, that is important to highlight.

We agree that the lasso is generally better for prediction rather than inference due to only selecting one of two equally good predictors to keep in the model at each stage. We considered applying the bootstrapped lasso (Bach 2008; arXiv:0804.1302), but had too few observations given the number of predictors. The covariance test for the lasso which we used instead has been shown to be more useful for inference (as cited), and we think still useful for finding potential lineage effects (which are collinear with the genetic background). To ensure robust analysis here, we also used a variance components analysis to test the variance explained and significance of all serotypes and resistances together.

However, we agree that the limitations of the lasso apply to the analysis mentioned in the subsection “Serotype and drug resistance explain part of the narrow-sense heritability” where we have attempted to find whether serotype and resistance have an effect independent of genetic background. As suggested, we have performed a more robust analysis with a linear mixed model, and found that only 19F was associated independent of background. We have highlighted these differences, and mentioned the drawbacks of the lasso approach in this section as suggested. Additionally, as part of our response to major point 2), we have also been more careful in describing association/causation in this section.

2) The authors need to be more careful/speculative in their description of their findings, especially in the Abstract. For example, with respect to the role of host factors, this was far from exhaustive in this study, and there are many more host features, that were the data available could have accounted for more of the phenotype variation, so I believe it is more accurate to state "We estimated that pneumococcal genomic variation accounted for 63% of the phenotype variation, whereas the host traits considered here accounted for less than 5%." (Clearly, the inclusion of epidemiological metadata, if any are available, would enrich the story.)

The second statement in the Abstract that is overstated is 'A pan-genome-wide association study identified prophage sequences as significantly decreasing carriage duration independent of serotype.' Whereas without any evidence of causation, I believe they can only state that they have "identified prophage sequences that significantly associated with decreasing carriage duration independent of serotype." The same idea holds for the association of erithromycin resistance and carriage – the causality is not straightforward. Under the theory of Lehtinen et al. (2017), long durations drive resistance due to the sensitivity of strains with long carriage durations to the fitness effects of antibiotic use, not the other way around.

We have revised the writing of the findings to ensure causality is not implied where we do not have evidence for it, and have been careful to describe associations appropriately. We have made the suggested changes to the Abstract, and similar changes to describing the effect of erythromycin resistance and the section testing for effects of capsule versus genetic background (as mentioned in the response to major point 1).

3) The association of the polymorphic phage with carriage is intriguing and deserves more treatment. Phages likely affect the strain in very different ways, some potentially extending carriage duration with other others decreasing it. Can phages in the database be partitioned into categories based on predicted phenotypic impact and the association retested? What was the genetic polymorphism in the phage genes that associated with carriage duration? How many strains was this in, i.e., was it a large effect in a small number of samples or a small affect in a large number? What was the mean carriage duration for strains with and without it (similar to that shown for serotype in Table 2). Phage often encode specific virulence or immune evasion factors, does this phage carry these? (Ideally a lab-based assay could be performed on the clinical strains to provide some functional information about its potential effect, or a second collection of isolates could be used to validate the finding, but neither is required.)

We have now conducted further investigation into the effect of prophage, which is reported in in the Results, as well as increasing the detail given around the previous section devoted to this analysis. An interesting new finding we report is that interruption of *comYC* by prophage is highly associated with reduced carriage duration, which may explain the association of phage sequence with carriage duration. We have added more detail on this result in the Discussion section.

These additions answer all the questions posed here, with two exceptions. Partitioning phages in the database based on predicted phenotypic impact: unfortunately, this is not possible, as we do not have the carriage duration for prophage in this database. We have not reported mean carriage duration with and without the phage in the same way as serotype: as only a subset of carriage episodes are swabbed this would lead to an estimate which was biased upwards. We however now include effect sizes and minor allele frequencies, and interpret these in the context of the *comYC* association.

Sadly, we do not have access to the clinical strains to perform an assay to validate the finding, but we are planning analysis of a similar study to be performed in Cape Town as noted, which should help validate these results in future (though two years of longitudinal sampling to determine carriage durations will be required).